# Synthesis and Characterization of Nanostructured Multi-Layer Cr/SnO_2_/NiO/Cr Coatings Prepared via E-Beam Evaporation Technique for Metal-Insulator-Insulator-Metal Diodes

**DOI:** 10.3390/ma15113906

**Published:** 2022-05-31

**Authors:** Sana Abrar, Muhammad Bilal Hanif, Abdulaziz Salem Alghamdi, Abdul Khaliq, K. S. Abdel Halim, Tayyab Subhani, Martin Motola, Abdul Faheem Khan

**Affiliations:** 1Department of Materials Science and Engineering, Institute of Space Technology, 1-National Highway, Islamabad 44000, Pakistan; sanaabrar46@gmail.com; 2Department of Inorganic Chemistry, Faculty of Natural Sciences, Comenius University in Bratislava, Ilkovicova 6, Mlynska Dolina, 842 15 Bratislava, Slovakia; hanif1@uniba.sk; 3College of Engineering, University of Ha’il, Ha’il P.O. Box 2440, Saudi Arabia; a.alghamdi@uoh.edu.sa (A.S.A.); ab.ismail@uoh.edu.sa (A.K.); k.abdulhalem@uoh.edu.sa (K.S.A.H.); 4Central Metallurgical Research and Development Institute (CMRDI), P.O. Box 87, Helwan 11421, Egypt

**Keywords:** metal-insulator-insulator-metal (MIIM), Rutherford backscattering, conductivity, I-V curves, Cr/SnO_2_/NiO/Cr coatings

## Abstract

Enhanced non-linearity and asymmetric behavior of the Cr/metal oxide diode is reported, with the addition of two insulator layers of SnO_2_ and NiO to form the metal-insulator-insulator-metal (MIIM) configuration. Such an MIIM diode shows potential for various applications (rectifiers and electronic equipment) which enable the femtosecond fast intoxication in MIIM diodes. In this work, nanostructured multi-layer Cr/SnO_2_/NiO/Cr coatings were fabricated via e-beam evaporation with the following thicknesses: 150 nm/20 nm/10 nm/150 nm. Coatings were characterized via Rutherford backscattering (RBS), scanning electron microscopy (SEM), and two-probe conductivity testing. RBS confirmed the layered structure and optimal stoichiometry of the coatings. A non-linear and asymmetric behavior at <1.5 V applied bias with the non-linearity maximum of 2.6 V^−1^ and the maximum sensitivity of 9.0 V^−1^ at the DC bias point was observed. The promising performance of the coating is due to two insulating layers which enables resonant tunneling and/or step-tunneling. Based on the properties, the present multi-layer coatings can be employed for MIIM application.

## 1. Introduction

Ever since their discovery, metal-insulator-metal (MIM) devices play a crucial role for a plethora of applications, including hot-electron high-speed diodes, large-area microelectronics, and tunneling cathodes [1,2,3]. The three significant parameters of MIM diodes are non-linearity, asymmetry, and turn-on voltage (V_on_). The three aspects are directly affected by the charge carrier transport efficiency through the MIM diode. Here, the type of the insulator is substantial for the MIM diode to be efficient [4,5,6,7]. For instance, oxide insulators with a wide bandgap (E_BG_) are limited due to their high V_on_ [8,9]. Therefore, insulators with a narrow E_BG_ are preferred (e.g., Nb_2_O_5_ and Ta_2_O_5_) as they possess, in general, a lower V_on_ value. Moreover, the presence of quantum tunneling will achieve high-speed, steady, and temperature-insensitive conduction [10,11].

Two critical factors are responsible for an ultra-fast diode for rectennas in an MIM diode, i.e., thermionic emissions and quantum mechanical tunneling. The thickness of the insulating coating is decisive for the efficient rectification. Indeed, the thinner the insulating coating is, the higher the current that is delivered [12,13], i.e., quantum tunneling provides the most efficient rectification with a <10 nm thickness of the layer [14]. Asymmetry of the MIM diode is directly responsible for this output, and two distinct electrodes with different work functions are necessary to achieve this physical state. For this, metal electrodes with high conductivity are often used [15], and the conductivity is further improved (to ultrafast response) due to quantum tunneling [1]. The current-voltage (I-V) asymmetry can be achieved by selecting suitable metals with high operating voltage for the overall MIM diode I-V response. Other factors affect the I-V response, including non-uniformity, outer ruggedness, and pinholes present on the surface of MIM diode. Therefore, careful processing of successive layers/coatings of metal-insulator-metal, respectively, is crucial for the final I-V response of the MIM diode. The commercialization of full electronics based on MIM is inhibited due to lack of repeatability (e.g., non-uniform coatings) and high yield system [15,16]. Metal-insulator interface roughness and oxide stoichiometry consumed the period of progress in this field. Nevertheless, by optimizing the fabrication of the MIM coatings, good conductivity and cutting-edge asymmetry were achieved [10].

By adding an additional insulator into the MIM (i.e., metal-insulator-insulator-metal, MIIM), progress in the non-linearity and asymmetry is achieved. MIIM diodes possess the ability to manipulate the trade-off between the resistance and responsitivity. To achieve this, the two insulating coatings must possess different volumes, flexibility, and electron affinity so different tunnel effects in the MIIM diodes is present [12]. In addition, phase tunneling is also present in MIIM diodes [1,16]. Indeed, phase tunneling results in substantial asymmetry, thus, improving the overall phase tunneling. 

In the present work, to produce the tunnel junction gadgets, a combination of a p-type NiO with an n-type SnO_2_ was explored. The combination of NiO/SnO_2_ coatings in the MIIM diode was selected due to the following: (i) at high frequencies the coating acts as a resonant phase tunneling diode and (ii) at low frequencies as a typical p-n junction phase tunnel diode [17,18]. The rectification capabilities of NiO/SnO_2_ have been shown to be appropriate for constructing efficient p-n and Schottky junctions [17,18]. The n-type charge carriers in the SnO_2_ layer are known to accumulate at the interface to form a sheet with high conductance, and the polar discontinuities at the interfaces aid in generating and/or removing the charge carriers [17]. As a metal, Cr was chosen as it is widely used for MIM and/or MIIM diodes due to its work function properties as an electrode. Moreover, by adding the NiO/SnO_2_ coating, the current asymmetry increases along with the overall reliability of the MIIM diode [17]. To obtain this, nanostructured multi-layered Cr/SnO_2_/NiO/Cr coatings were fabricated via e-beam evaporation [19], and a thorough characterization including Rutherford backscattering (RBS) and scanning electron microscopy (SEM) was conducted. Conductivity and I-V characteristics were measured to confirm the use of the coatings for application as MIIM tunnel diodes.

## 2. Materials and Methods

The electron beam (e-beam) evaporation technique was used for the fabrication of nanostructured multi-layered Cr/SnO_2_/NiO/Cr coatings using the following precursors: Cr (99.99%, MERK, Missouri, USA), SnO_2_ (99.99%, MERK, Gladstone, MO, USA), and NiO (99.99%, GMA-ALDRICH, Saint Louis, MO, USA). Coatings were evaporated on soda-lime glass at room temperature. Briefly, a 1 g pellet (Cr, SnO_2_, NiO, respectively) with a thickness of 10 mm was prepared via a hydraulic press. Each pellet contained 0.95 g of precursor and 0.05 g of polyvinyl alcohol (PVA) gel as a binder. To lower the intrinsically high viscosity of PVA, deionized water was added to PVA and the solution was stirred for 3 h at 90 °C. Subsequently, the pellets were placed in a crucible (Mo crucibles were used to avoid any contamination) under a high vacuum (2 × 10^−6^ mbar) along with the soda-lime glass. Prior to the deposition, pre-evaporation was carried out for 60 s to remove any native surface oxides [20]. 

For elemental composition and depth profiling, Rutherford backscattering (RBS) spectroscopy was conducted with the 5 MeV Pelletron Tandem Accelerator (5UDH-2 pelletron, NEC, Irving, TX, USA). The particles (with an average energy of 2 MeV) with Cornell geometry were used to obtain the RBS spectrum and simulated using SIMNRA (V6.05 software, Dr. Matej Mayer, Max-Planck Institut fur Plasmaphysik, Garcing, Garching bei München, Germany). The surface morphology was analyzed via field emission scanning electron microscopy (FESEM, MIRA TESCAN, Milano, Italy) with an average accelerating voltage of 20 kV. According to our previous reports [21,22,23,24,25,26,27], current-voltage (I-V) was recorded by a DC-2-point probe (Keithley source meter 2600 series) using Ag paste with a mesh-like morphology as a current collector at room temperature on the MIIM device coating. As shown in Figure 1, both the contacts were made on the surface of a 50 × 50 mm sample. The I-V characteristic values were obtained by cycling the potential from −3 V to +3 V with a step rate of 0.1 V/s. The differences in the measurements of conductivity did not exceed ±5 %. Structural analysis of the multi-layer Cr/SnO_2_/NiO/Cr coating was conducted via X-ray diffractometer (XRD, Bruker D8 Discover, Bremen, Germany) equipped with Cu Kα in the 2θ range from 10° to 80° at room temperature.

## 3. Results and Discussion

All coatings (i.e., Cr; Cr/SnO_2_; Cr/SnO_2_/NiO; Cr/SnO_2_/NiO/Cr) deposited on soda-lime glass were physically stable, i.e., with a good adhesion to the underlying substrate. A simple tape test confirmed that the films were adherent with the substrates and no peel-off, delamination, cracks, or blister were found on the films (as shown in Figure 2).

Schematics, illustration photographs, and the coatings are shown in Figure 2. The following sequence was conducted to prepare the final nanostructured multi-layered Cr/SnO_2_/NiO/Cr coating. First, the Cr coating (Figure 2a; ~150 nm thickness, 0.2 nm/s deposition rate) was deposited on a glass substrate. Subsequently, SnO_2_ (Figure 2b; ~20 nm, 0.44 nm/s deposition rate), NiO (Figure 2c; ~150 nm thickness, 0.1 nm/s deposition rate), and Cr (Figure 2d; ~150 nm thickness, 0.2 nm/s deposition rate) were deposited, respectively.

Figure 3 shows the typical XRD pattern of multi-layer Cr/SnO_2_/NiO/Cr films (only one sample was performed due to limited availability of machine and time constraints). It can be observed that the figure contains only one broad band/halo, which suggests that the present films are amorphous in nature. Furthermore, it may also be concluded from the XRD pattern that the crystallite size might be in the nanoscale range.

Representative FESEM images of the coatings are shown in Figure 4. All coatings possess dense and uniform surfaces with no visible cracks or pinholes and good adhesion (determined by a tape test method) to the underlying substrate. Moreover, the figures illustrate that the layers consist of nanocrystals, which are spherical in nature. 

The experimental and fitted RBS spectra of the prepared coatings are shown in Figure 5. For the mono-layer Cr, a sole single peak representing Cr was determined in the RBS spectra (Figure 5a). The intended thickness (150 nm, theoretical calculations) of the Cr coating was not achieved but an increase in thickness to ~190 nm (experimental) was determined. This is due to the limitations of crystal quartz monitoring during deposition [28]. Elemental concentration analysis determined 60 wt.% of Cr and 40 wt.% of O. The relatively high amount of O stems from the underlying substrate. In the bi-layer Cr/SnO_2_ coating (Figure 5b), two major peaks appeared. The Sn peak appeared at a higher channel compared to that of Cr. This is due to the increased kinematic factor for Sn (0.9677) compared to that of Cr (0.9259). The two peaks represent the formation of a bi-layer structure in the Cr/SnO_2_ coating, as illustrated in Figure 2b. The thickness of ~190 nm and ~20 nm was obtained for Cr and SnO_2_. The RBS spectrum of the tri-layer Cr/SnO_2_/NiO coating (thickness ~190/~20/~10 nm) is shown in Figure 5c. Three major peaks, from the higher to the lower channel, represent Sn (kinematic factor 0.9677), Ni (kinematic factor 0.9341), and Cr (kinematic factor 0.9259), respectively. At last, RBS spectra of the tetra-layer Cr/SnO_2_/NiO/Cr coating (thickness ~190/~20/~10/~150 nm) are shown in Figure 4d. The spectrum of the tetra-layer is similar to that of the tri-layer (Figure 4c) due to the similar composition of the coatings. All in all, thickness, stoichiometry, and layered structure was obtained in all coatings, which is crucial for application in MIIM.

Electrical properties were measured through a 2-point probe technique and the resistance and conductivity were subsequently calculated using the following equations (Equations (1) and (2)) [29,30,31,32,33,34,35]: ρ (resistivity) = R (resistance) × t (thickness)(1)
σ (conductivity) = 1/ρ(2)

The conductivity of the coatings is depicted in Figure 6. The conductivity of the mono-layer Cr coating was 7.45 × 10^4^ (Ω-cm)^−1^, which is similar to that of pure Cr with a thickness of 200 nm (6.4 × 10^4^ (Ω-cm)^−1^ and 400 nm (6.8 × 10^4^ (Ω-cm)^−1^), respectively. In the present study, the conductivity is slightly higher, which may be due to the smooth and uniform coating, which is already represented by RBS analysis and SEM micrographs as compared to previous research [36,37]. Reduced conductivity was observed in the bi-layer Cr/SnO_2_ coating 2.88 × 10^−1^ (Ω-cm)^−1^. This is due to the intrinsically low conductivity of SnO_2_ compared to that of Cr. In general, SnO_2_ coatings with thicknesses between 60 and 300 nm possess conductivity of 4.3 × 10^−1^ (Ω-cm)^−1^ and are further reduced in thinner coatings. A slight increase in conductivity to 8.84 × 10^−1^ (Ω-cm)^−1^ was observed in the tri-layer Cr/SnO_2_/NiO due to the addition of NiO. Indeed, thin NiO coatings (>10 nm) possess a conductivity of 0.01063 × 10^−3^ (Ω-m)^−1^. In the tetra-layer Cr/SnO_2_/NiO/Cr, a substantial increase in the conductivity to 2.617 × 10^3^ (Ω-cm)^−1^ was determined. The sudden increase in conductivity is due to the metallic nature of Cr. Indeed, E_BG_ of Cr is narrow (approaches to zero) and the electrons can efficiently move from the valence to the conduction band. Nevertheless, several aspects need to be considered when discussing the overall conductivity of our MIIM device (i.e., tetra-layer Cr/SnO_2_/NiO/Cr). In general, several aspects affect the conductivity of material including temperature and the nature of a semiconductor. Besides these crucial parameters, the thickness and surface roughness of the coating is also an important factor. With decreasing surface roughness, the smoothness of the coating increases with defects and strains variations which hinder the transport of electrons, i.e., increasing the resistivity and decreasing the conductivity. On the other hand, the thicker the coatings are, the less conductive it is. These bold statements are confirmed by our experimental data which showed a decrease in conductivity of the tetra-layer Cr/SnO_2_/NiO/Cr compared to that of the mono-layer Cr. This is due to the scattering of electrons at the Cr/SnO_2_/NiO/Cr interfaces where two insulating coatings (SnO_2_ and NiO) are placed between the two metal ones (Cr and Cr).

At last, I-V characteristics of the tetra-layer Cr/SnO_2_/NiO/Cr coatings were evaluated. As depicted in Figure 7, non-linear and asymmetric behavior at <1.5 V applied bias with the non-linearity maximum of 2.6 V^−1^ and the maximum sensitivity of 9.0 V^−1^ at the DC bias point was observed. In general, it is crucial for MIIM diodes to possess a substantial non-linearity in order to obtain a significant response. As widely reported [38,39,40], maximum sensitivity reached 5 V^−1^ at >1 V. Our presented design of the MIIM diodes enhanced its sensitivity by approximately 1.8 times and the reason for that is the following. By utilization of the metal electrodes with a work function difference provides increased non-linearity. It indicates that the conduction mechanism is due to the electron tunneling through the thin insulating layer (here, the 20 nm and 10 nm thick SnO_2_ and NiO, respectively). By adding two different insulating layers between the two metal layers (MIIM), a substantial performance enhancement was achieved compared to the widely used MIM. The rectifying characteristics of the MIIM diode containing SnO_2_, NiO, and Cr were analyzed in terms of linearity and asymmetry, and a defined turn-on toltage of the current-voltage (IV) measurements. According to these characteristics, the performance of the relevant MIIM diode based on Cr/SnO_2_/NiO/Cr coatings was assessed. The reasons for this optimal composition of the coatings is described in the Introduction, i.e., each of the components possess unique and suitable properties for the overall efficiency of the diode. Therefore, the presented tetra-layer Cr/SnO_2_/NiO/Cr coating is efficient for applications in MIIM diodes.

## 4. Conclusions

In summary, we report the fabrication of a nanostructured multi-layer Cr/SnO_2_/NiO/Cr MIIM diode and its I-V characterization. Smooth, dense, and uniform coatings were obtained via e-beam evaporation with good control of thickness that showed suitable asymmetry and non-linearity of the MIIM diode, i.e., non-linear and asymmetric behavior at <1.5 V applied bias with the non-linearity maximum of 2.6 V^−1^. The maximum sensitivity of the MIIM diode was 9.0 V^−1^ at the DC bias point. The resonant tunneling (quantum tunneling) dominated via the trap-assisted tunneling mechanism in the four-layer dielectric diode. In addition, a conductivity of 2.88 × 10^3^ (Ω-cm)^−1^ was obtained, which is optimal for MIIM diodes. The conductivity decreased from the initial value of 7.74 × 10^4^ (Ω-cm)^−1^ for Cr due to higher scattering of electrons due to two different insulating layers (SnO_2_ and NiO) and with an increased overall thickness of the coating. All in all, we have shown that our designed nanostructured Cr/SnO_2_/NiO/Cr coatings as an MIIM diode possess great potential for future rectification devices.

## Figures and Tables

**Figure 1 materials-15-03906-f001:**
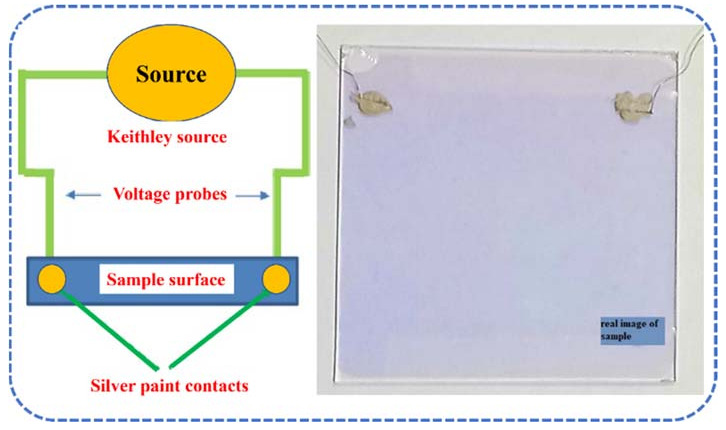
Schematic of two-probe conductivity method with real image of the sample.

**Figure 2 materials-15-03906-f002:**
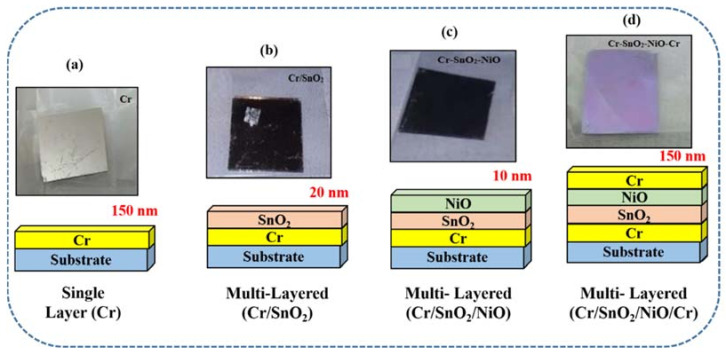
Schematic representation and illustrative photographs of (**a**) Cr, (**b**), Cr/SnO_2_, (**c**) Cr/SnO_2_/NiO, and (**d**) Cr/SnO_2_/NiO/Cr coatings, respectively.

**Figure 3 materials-15-03906-f003:**
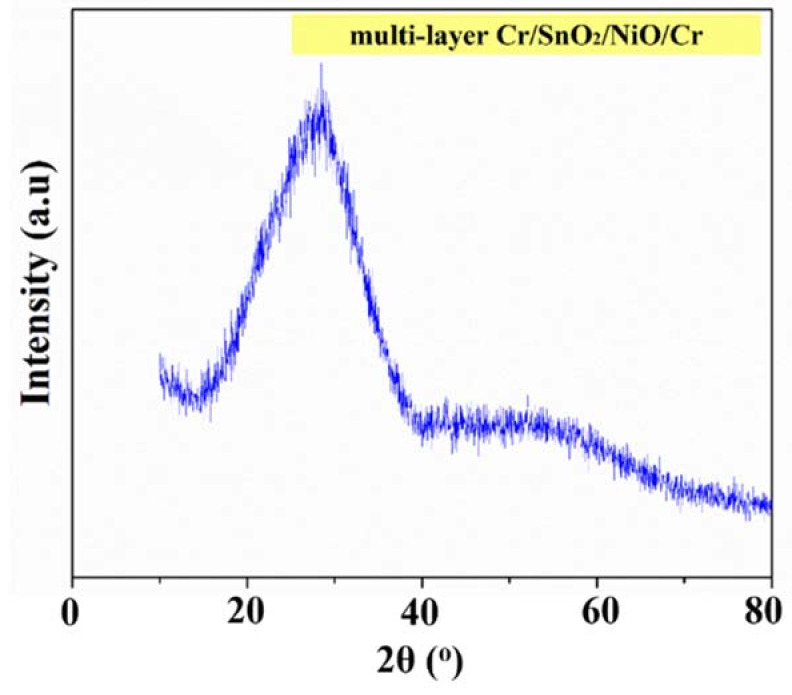
XRD pattern of the multi-layer Cr/SnO_2_/NiO/Cr coatings.

**Figure 4 materials-15-03906-f004:**
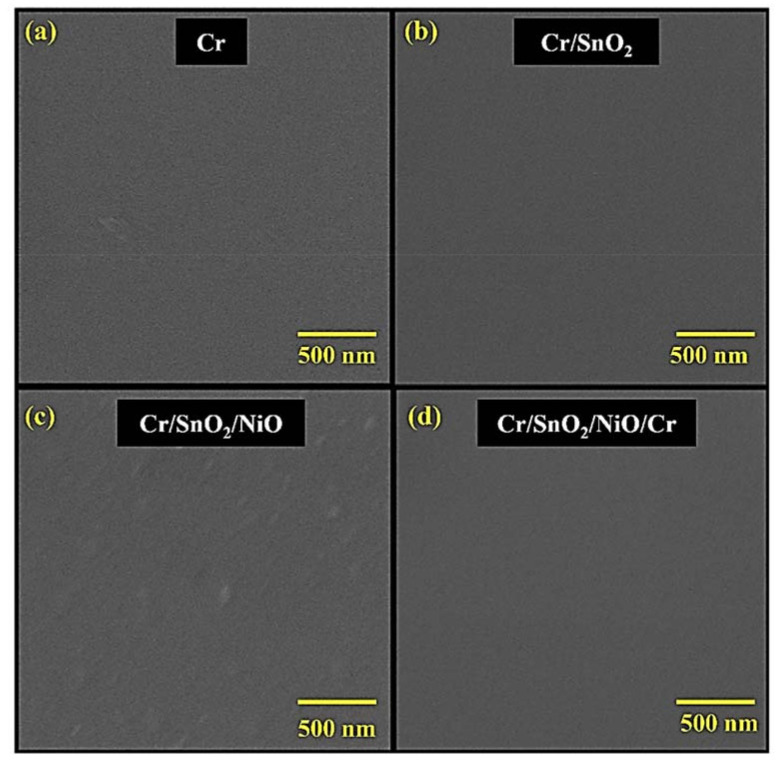
FESEM images of (**a**) Cr, (**b**), Cr/SnO_2_, (**c**) Cr/SnO_2_/NiO, and (**d**) Cr/SnO_2_/NiO/Cr coatings, respectively.

**Figure 5 materials-15-03906-f005:**
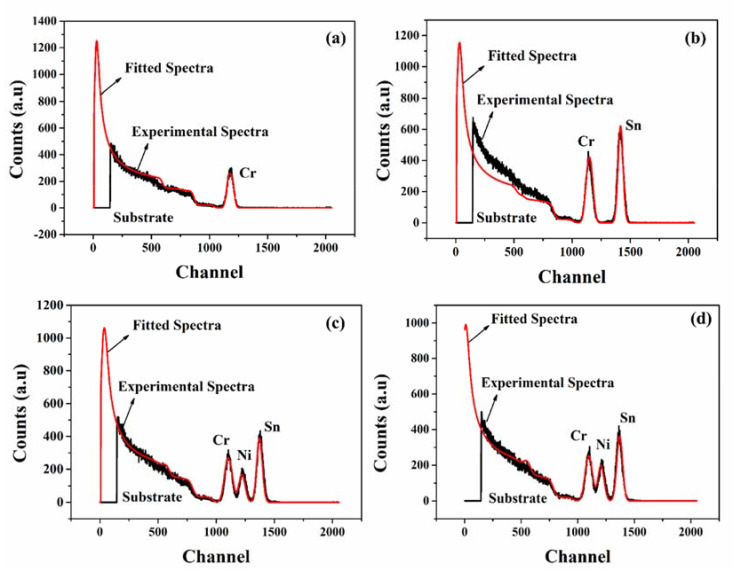
Experimental and fitted RBS spectra of (**a**) Cr, (**b**), Cr/SnO_2_, (**c**) Cr/SnO_2_/NiO, and (**d**) Cr/SnO_2_/NiO/Cr coatings, respectively.

**Figure 6 materials-15-03906-f006:**
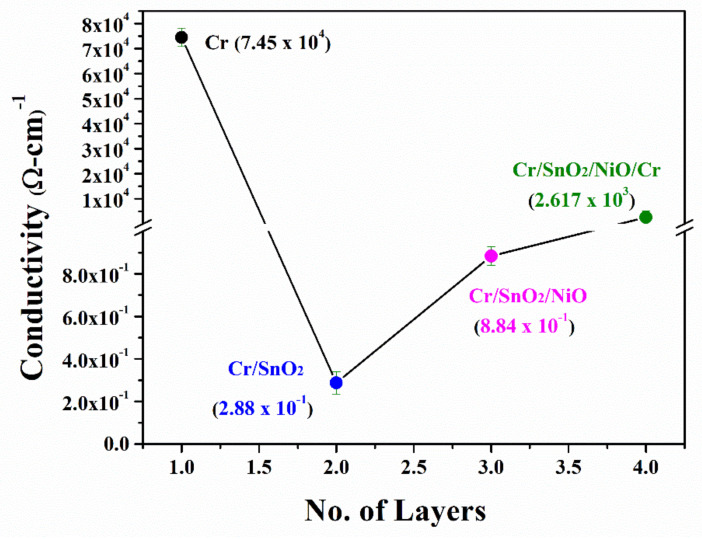
Electrical conductivity of Cr, Cr/SnO_2_, Cr/SnO_2_/NiO, and Cr/SnO_2_/NiO/Cr coatings, respectively.

**Figure 7 materials-15-03906-f007:**
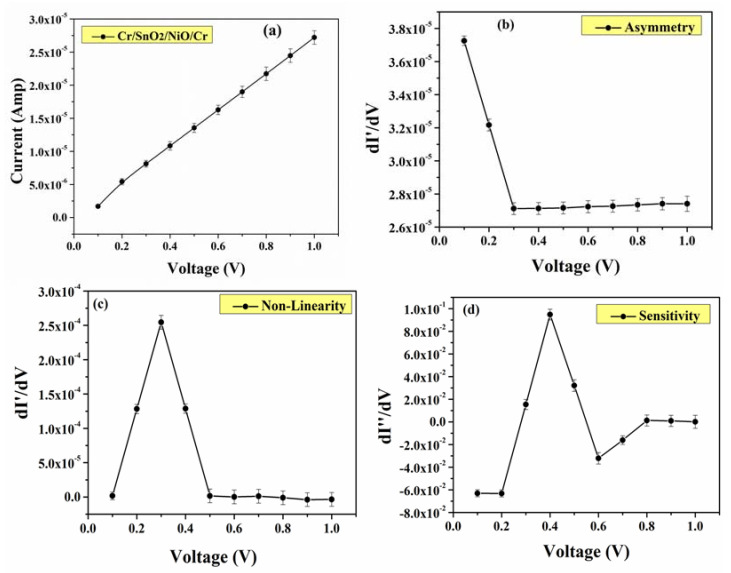
I-V characteristics of MIIM device (Cr/SnO_2_/NiO/Cr) (**a**). Asymmetry (**b**), Non-linearity (**c**) and Sensitivity (**d**) was derived from I-V values.

## Data Availability

The data presented in this study are available upon request from the corresponding author.

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
