# Peer review of "Synthesis and Characterization of Nanostructured Multi-Layer Cr/SnO2/NiO/Cr Coatings Prepared via E-Beam Evaporation Technique for Metal-Insulator-Insulator-Metal Diodes"

_materials, 2022, doi:10.3390/ma15113906_

Round 1
Reviewer 1 Report
The paper entitled 'Synthesis and characterization of nanostructured multi-layer Cr/SnO2/NiO/Cr coatings prepared via e-beam evaporation technique' by Sana Abrar et al. present the possible application of Cr/SnO2/NiO/Cr multilayer coating in Metal-insulator-insulator-metal application. The work is well written and easy to understand. Thus I suggest publication.
Author Response
Response to Reviewer #1: The paper entitled 'Synthesis and characterization of nanostructured multi-layer Cr/SnO2/NiO/Cr coatings prepared via e-beam evaporation technique' by Sana Abrar et al. present the possible application of Cr/SnO2/NiO/Cr multilayer coating in Metal-insulator-insulator-metal application. The work is well written and easy to understand. Thus, I suggest publication.
Response: Dear Reviewer, I really appreciate the expertise of the reviewer and again thank you for your acceptance of the paper in its present form.

Reviewer 2 Report
The paper ”Synthesis and characterization of nanostructured multi-layer Cr/SnO2/NiO/Cr coatings prepared via e-beam evaporation technique” is suitable for publication in Materials Journal, after some minor corrections. The introduction is well written with specific information in the field of MIM devices and presents the characteristics of different types of coatings. In my opinion, the authors should improve the quality/contrast of the figure 2-is not very clear despite the high magnitude. Authors should add XRD analysis in order to have a complimentary discussion between SEM analysis and phase compound identification. Should be added the ICDD files.
In rest is ok!
Author Response
Response to Reviewer #2: The paper “Synthesis and characterization of nanostructured multi-layer Cr/SnO2/NiO/Cr coatings prepared via e-beam evaporation technique” is suitable for publication in Materials Journal, after some minor corrections. The introduction is well written with specific information in the field of MIM devices and presents the characteristics of different types of coatings.
In my opinion, the authors should improve the quality/contrast of figure 2 is not very clear despite the high magnitude. Authors should add XRD analysis in order to have a complimentary discussion between SEM analysis and phase compound identification. Should be added the ICDD files.
Response: Dear Reviewer, thank you for your kind comments and suggestions. Figure. 2 has been changed, and the XRD results have been added to the revised version.

Reviewer 3 Report
The manuscript titled "Synthesis and characterization of nanostructured multi-layer Cr/SnO2/NiO/Cr coatings prepared via e-beam evaporation technique" proposes fabrication method to achieve metal-insulator-metal structure usable for application as a tunneling diode material. However, the experiments are poorly designed, the results are not clearly presented, and the research topic is described inadequately. The introduction is hardly intelligible, the research topic is not described clearly, particular sentences, such as: "Polar system gravitational anomalies contribute to the development or depletion of charging companies." do not make any sense at all. The experimental section descibes weirdly in detail the pellet preparation or the material evaporation, then there is crucial information missing about the layout of the resistivity and I-V characteristics measurements. The SEM images show hardly any significant results, maybe FIB-SEM images (or simle scratch test) of the sample cross-section are in place to evaluate the layered sample structure. RBS fail to show much significant data as well, save the mere fact that Cr, Sn and Ni are present in the samples. The detail on how the oxygen amount was estimated (elemental concentration analysis) is missing completely. The worst part are the electrical properties measurements. The nanometer-thick layers very likely showed extremely low values of resistance (not sure, data is missing). Therefore, the 2-point probe is just insufficient to measure the resistance values properly, since the resistance of the measurement probes themselves influences the values of measured resistance significantly. The information on the layout of the electric measurements is crucial to evaluate, if the measurements were even conducted properly (the placement of the measurement electrodes). The equation for calculation of resistivity is just wrong, if you are trying to measure the resistivity over the structure thickness. Moreover, the presented deviations of the I-V characteristics from linearity are minimal, therefore the rectification efficiency of such a structure would probably be non-existent (no evaluation of this is even present in the work, or of rectification efficiency dependence on signal frequency).
Author Response
Response to Reviewer #3: The manuscript titled "Synthesis and characterization of nanostructured multi-layer Cr/SnO2/NiO/Cr coatings prepared via e-beam evaporation technique" proposes a fabrication method to achieve a metal-insulator-metal structure usable for application as a tunneling diode material. However, the experiments are poorly designed, the results are not clearly presented, and the research topic is described inadequately. The introduction is hardly intelligible, the research topic is not described clearly, and particular sentences, such as: "Polar system gravitational anomalies contribute to the development or depletion of charging companies." do not make any sense at all. The experimental section describes weirdly in detail the pellet preparation or the material evaporation, then there is crucial information missing about the layout of the resistivity and I-V characteristics measurements. The SEM images show hardly any significant results, maybe FIB-SEM images (or simple scratch test) of the sample cross-section are in place to evaluate the layered sample structure. RBS fails to show many significant data as well, save the mere fact that Cr, Sn, and Ni are present in the samples. The detail on how the oxygen amount was estimated (elemental concentration analysis) is missing completely. The worst part is the electrical properties measurements. The nanometer-thick layers very likely showed extremely low values of resistance (not sure, data is missing). Therefore, the 2-point probe is just insufficient to measure the resistance values properly, since the resistance of the measurement probes themselves influences the values of measured resistance significantly. The information on the layout of the electric measurements is crucial to evaluate if the measurements were even conducted properly (the placement of the measurement electrodes). The equation for the calculation of resistivity is just wrong if you are trying to measure the resistivity over the structure thickness. Moreover, the presented deviations of the I-V characteristics from linearity are minimal, therefore the rectification efficiency of such a structure would probably be non-existent (no evaluation of this is even present in the work, or of rectification efficiency dependence on signal frequency).
Response: Dear Reviewer, thank you so much for your detailed analysis and for valid suggestions. We address them as follows:
- In the revised version of the manuscript, we added XRD (Figure 2) to improve the quality of the work.
- Regarding FIB-SEM we actually agree with the reviewer that these data would provide additional information regarding the layered structure. However, the FIB-SEM has its limitation for cross-sectional imaging. The samples were first tried to measure the thickness through cross-sectional imaging using FIB-SEM but it did not provide adequate results for the few nanometer thickness coating on a glass substrate. Nevertheless, the layered structure is confirmed by RBS which is, for this purpose a more valid information than FIB-SEM. The different layers are clearly distinguished by RBS so we believe that there is no need for FIB-SEM. Although we’d obtain some good-looking SEM images, they would in no way somehow improve the quality of the presented results. The layered structure is also confirmed by changes in the conductivity as discussed in the original version of the manuscript. So, considering all the presented results, the layered structure of our MIIM device is obvious.
- The RBS technique has been considered very worthy at the nanoscale level to find out the composition and thickness of the nano-scale films/coatings. Furthermore, it has an additional advantage of measuring the quantitative compositional depth profile of 0.1 nm thick films. The accuracy of measuring oxygen in RBS is far better than in SEM, which has its limitation for lighter elements. In the present case, the concentration of oxygen in coatings was estimated via RBS. This estimation is ordinary as RBS is often used to determine the amount of oxygen (e.g., https://doi.org/10.1080/15361055.2021.1927624).
- We agree that the conductivity measurements conducted by the 2-probe system does not provide a comprehensive analysis of conductivity, however, it is generally accepted by the scientific community. In the revised version of the manuscript, we added more detailed information about the conductivity measurements, and we also cite additional papers from our group where the 2-probe system was used to determine the conductivity of different types of layers.
- At last, we completely revised the Introduction to make it clearer and more understandable.

Reviewer 4 Report
This manuscript discussed about one 4 layer nanocoating material through e-beam evaporation, the synthesis method and some characterization were performed. It is a traditional work but also has some research potentials. So I would still suggest to publish after these questions and comments are addressed.
(1) First some figures need to be adjusted, I don't know the reason but the resolution quality for many figures is not that good. Like figure 1, words are blurry, photos are not in good quality, which I believe it is easy to improve. The authors have tendency to use red as the word color in figures, which is not often seen in research papers. Also Figure 5 and 6, fonts are too small to read.
(2) The major point of this manuscript is the material coating itself. So authors need to explain why you need to coat these 4 specific layers, also various thickness and deposition rate is also critical. I feel introduction is not enough, only the final paragraph talks about the material you would like to coat. But the reason might not be enough, need more references to reinforce the importance for this coating.
(3) In abstract, introduction, discussion, all need to explain the reasons for these 4 layers coating and why it is beneficial. Why authors would like to work on this 4 layer coating research area needs to be explained well.
(4) Personally I feel figure 3 is not useful, but it is ok. Figure 4 also did not see some differences. How does authors figure out the thickness of these coatings?
(5) Conclusion can be somehow improved and now it still looks like data summary report, I suggest talking about major points and point out potential areas and future work.
Author Response
Dear Reviewer:
Find attached comments from all the reviewers.

Round 2
Reviewer 3 Report
The introduction section was improved moderately, it is still far from describing the studied problem comprehensibly. Authors state they used RBS for depth profiling, yet there is no depth profile graph of the samples in the manuscript, no discussion of the RBS spectra, the graphs in the manuscript just show, there is Cr, Sn and Ni present in the samples. The XRD spectrum added in the revision has no value for the work. I am still not sure what is the position of the Ag contacts for the I-V measuremetns. If it means one of the contacts is on the back of the (insulating) substrate, the measurements have no value, since the electrical propertiess will be governed by the parameters of the substrate material, which is orders of magnitude thicker than the prepared structure itself. The most important part of the work is therefore flawed on a very basic fundamental level and the work just should not be published.
Author Response
Reviewers’ Comments
Response to Reviewer #1: The paper entitled 'Synthesis and characterization of nanostructured multi-layer Cr/SnO2/NiO/Cr coatings prepared via e-beam evaporation technique' by Sana Abrar et al. present the possible application of Cr/SnO2/NiO/Cr multilayer coating in Metal-insulator-insulator-metal application. The work is well written and easy to understand. Thus, I suggest publication.
Response: Dear Reviewer, I really appreciate the expertise of the reviewer and again thank you for your acceptance of the paper in its present form.
Response to Reviewer #2: The paper “Synthesis and characterization of nanostructured multi-layer Cr/SnO2/NiO/Cr coatings prepared via e-beam evaporation technique” is suitable for publication in Materials Journal. The authors addressed all the comments.
Response: Dear Reviewer, thank you so much for your acceptance of the paper.
Response to Reviewer #3: The introduction section was improved moderately, it is still far from describing the studied problem comprehensibly. The authors state they used RBS for depth profiling, yet there is no depth profile graph of the samples in the manuscript, no discussion of the RBS spectra, the graphs in the manuscript just show, there is Cr, Sn, and Ni present in the samples. The XRD spectrum added in the revision has no value for the work. I am still not sure what is the position of the Ag contacts for the I-V measurements. If it means one of the contacts is on the back of the (insulating) substrate, the measurements have no value, since the electrical properties will be governed by the parameters of the substrate material, which is orders of magnitude thicker than the prepared structure itself. The most important part of the work is therefore flawed on a very basic fundamental level and the work just should not be published.
Response: Dear Reviewer, thank you so much for your comments and suggestions
With due respect, the RBS depth profiling was stated due to the significance of the ion beam technique over FIB-SEM. We have never mentioned that we have done depth profiling in this manuscript. Still, the XRD pattern has much significance in the sense that, now we are assured that the deposited multi-layer films are amorphous rather than we assume them crystalline.
The IV characteristic has now been elaborated with a schematic and real picture of the sample as shown in Fig. 1. The silver contacts were made on the surface of the sample and not on the backside of the sample. As it is very obvious that the substrate is a glass in this study, which is completely insulator, there is no sense in making the back contact. In this case, as you mentioned, we will not get any IV value.
Reviewer 4 Report
This manuscript has gone through careful revision and fulfilled my requirement. The authors could pay more attention to the research goal and mechanism in the future work. I suggest this manuscript to be published in the present form.
Author Response
Reviewers’ Comments
Response to Reviewer #1: The paper entitled 'Synthesis and characterization of nanostructured multi-layer Cr/SnO2/NiO/Cr coatings prepared via e-beam evaporation technique' by Sana Abrar et al. present the possible application of Cr/SnO2/NiO/Cr multilayer coating in Metal-insulator-insulator-metal application. The work is well written and easy to understand. Thus, I suggest publication.
Response: Dear Reviewer, I really appreciate the expertise of the reviewer and again thank you for your acceptance of the paper in its present form.
Response to Reviewer #2: The paper “Synthesis and characterization of nanostructured multi-layer Cr/SnO2/NiO/Cr coatings prepared via e-beam evaporation technique” is suitable for publication in Materials Journal. The authors addressed all the comments.
Response: Dear Reviewer, thank you so much for your acceptance of the paper.
Response to Reviewer #3: - Error/uncertainty bars must be added to the results in Figs. 6 and 7.
Response: Dear Reviewer, I really appreciate the expertise of the reviewer and again thank you for your acceptance of the paper in its present form.
We revised Figures 6 and 7 according to the reviewer’s suggestion.
We appreciate the positive feedback on our work. We addressed all the changes in the revised version of the manuscript.